# Quantification and visualization of US racial geography using the National Racial Geography Dataset 2020

**Anna Dmowska**[1]⊕*, **Tomasz F. Stepinski**[2]⊕

**1** Institute of Geoecology and Geoinformation, Adam Mickiewicz University, Poznan, Poland, **2** Space Informatics Lab, Department of Geography and GIS, University of Cincinnati, Cincinnati, OH, United States of America

⊕ All these authors are contributed equally to this work.
* dmowska@amu.edu.pl

**Data Availability Statement:** Data are available in the NRGD2020 section at https://socscape.edu.pl website.

**Funding:** This project was supported by the National Science Centre, Poland (Narodowe

## Abstract

Racial geography studies the spatial distributions of multiracial populations. Technical challenges arise from the fact that US Census data, upon which all US-based studies rely, is only available in the form of spatial aggregates at a few levels of granularity. This negatively affects spatial analysis and, consequently, the quantification of racial segregation, especially on a smaller length scale. A recent methodology called the Racial Landscape (RL) stochastically disaggregates racial data at the level of census block aggregates into a grid of monoracial cells. RL-transformed racial data makes possible pattern-based, zoneless analysis, and visualization of racial geography. Here, we introduce the National Racial Geography Dataset 2020 (NRGD2020)—a collection of RL-based grids calculated from the 2020 census data and covering the entire conterminous US. It includes a virtual image layer for a bird's-eye-like view visualization of the spatial distribution of racial sub-populations, numerical grids for calculating racial diversity and segregation within user-defined regions, and pre-calculated maps of racial diversity and segregation on various length scales. NRGD2020 aims to facilitate and extend spatial analyses of racial geography and to make it more interpretable by tightly integrating quantitative analysis with visualization (mapping).

## Introduction

According to the US Census Bureau (www.census.gov), the population of the United States in 2020 was approximately 331 million people. Among this population, approximately 70% identified as White, around 19% identified as Hispanic, about 12% identified as Black, and roughly 6% identified as Asian. This makes the US racially diverse. Each of these racial sub-populations has its own spatial distribution. If these distributions were similar, the population would be racially integrated; however, in the US, racial sub-populations are distributed differently, leading to the phenomenon of racial segregation [1].

Segregation and diversity are the two most prominent features of racial geography. Racial geography, as defined by Huiping et al. [2], pertains to the residential distribution of the

Centrum Nauki), Agreement No. 2022/06/X/HS4/00449 URL: https://www.ncn.gov.pl/ The funders did not and will not have a role in study design, data collection and analysis, decision to publish, or preparation of the manuscript.

**Competing interests:** The authors have declared that no competing interests exist.

population based on self-identified race. The data collected during the 2020 US decennial census provides a comprehensive snapshot of the racial spatial patterns across the US. However, the spatial ambiguity of its aggregated format (introduced for privacy reasons) and the arbitrariness of the boundaries of aggregation areas impede spatial analysis.

On the quantitative front, assessing segregation of a region from aggregated data can only be done indirectly by comparing the diversity of this region (which must be an aggregation area) to the diversities of the constituting subsets of this region (which must be smaller aggregated areas). This is reflected in the construction of the most widely used segregation index—the information theory index $H$ [3, 4]. On the qualitative front, visualizing the spatial variability of segregation can only be done on the large length scale (referred to here for brevity as scale) by coloring the segregation of multiple regions (most frequently, counties) by their values of $H$.

The existing literature primarily focuses on calculating diversity and segregation indices for samples of large metropolitan areas [5–13]. This is not surprising, inasmuch as this analysis is supported by the aggregated census data. Additionally, such emphasis may also be attributed to the fact that much of the literature originates from the sociology community, primarily concerned with examining the connection between racial distribution and issues of discrimination and social justice. In this line of research, maps, if any, are typically created by assigning attributes such as racial composition or diversity metrics to the aggregated units (see, for example, Social Explorer maps at https://www.socialexplorer.com/) or by classifying census tracts based on combinations of racial attributes [14–16].

The utilization of more advanced spatial techniques in the field of racial research is rather rare and can be categorized into three distinct research directions. (1) The transformation of irregular census aggregation areas into a regular aggregation grid for easier analysis [17–20]. (2) The creation of one-person-per-dot maps for visualization (but not quantitative analysis) of racial distribution [21, 22]. (3) The regionalization of a region into sub-regions with more homogeneous racial characteristics [23–25].

A significant alternative approach to the analysis and visualization of racial geography was proposed by Dmowska et al. [26]. The key idea of this approach (called the Racial Landscape or RL for short) is to transform census block-level aggregated data into a *monoracial* grid. Each cell in the grid groups only inhabitants of the same race rather than inhabitants of several races, as in the census data and all previously proposed grids. For an in-depth description of the RL grid, refer to the "Materials and methods" section.

A gridded, non-aggregated dataset supports a pattern-based, zoneless analysis of racial geography, including the calculation of diversity and segregation metrics for an arbitrarily defined area. It also lends itself to meaningful mapping of racial geography, tightly integrated with quantitative metrics, as it is based on the same grid used for the calculation of metrics.

The RL method offers many advantages over the traditional method. However, it is new and different, which presents a barrier to its broader adoption. In this paper, we introduce the National Racial Geography Dataset 2020 (NRGD2020), a collection of geospatial layers calculated for the entire conterminous US using the RL method and utilizing data from the 2020 US decennial census. The aim of the NRGD2020 is to present the RL method in the form of ready-to-use data rather than a theoretical concept, thus lowering the barrier to its adoption. The choice of the name NRGD is deliberate, drawing inspiration from the National Land Cover Dataset or NLCD [27]. Our vision for the NRGD is to serve as a resource in demography, similar to how the NLCD is utilized in the field of land science.

The paper is organized as follows. In the section "Materials and methods", we provide a brief overview of the RL method and describe the construction of NRGD2020. In the section "Results", we list all layers constituting the NRGD2020 and discuss their meaning and purpose.

Section "Use cases" presents two use cases—the real problems examined in previous literature revisited using the NRGD2020. The last section is the summary and discussion.

## Materials and methods

In this section, we first provide a brief overview of the RL methodology. The full technical description of the RL methodology is given in the paper by Dmowska et al. [26]. Here, we provide only the information necessary for understanding the principles of the RL, mostly in an illustrative manner. Next, we describe a computational pipeline for the construction of NRGD2020.

### Racial Landscape brief

There are two key concepts behind the RL method: the first is the conversion of census block-level data to a monoracial grid, and the second is the zoneless calculation of segregation metrics from this grid.

A data conversion is schematically shown in the top row of Fig 1. Consider a hypothetical block with 27 inhabitants belonging to three different races and extending over a square area. Fig 1A underscores that block-based census data is aggregated and non-spatial; only the racial composition of the block's population is given. The RL divides the block's area into equal-sized cells; the hypothetical block happens to be divided into nine cells. Without any information about the internal distribution of inhabitants within a block, it is reasonable to assume a homogeneous distribution of each race (Fig 1B). However, such a choice results in multiracial cells. Thus, the RL distributes inhabitants stochastically into the cells in such a way that each cell is monoracial (Fig 1C). In this example, the population density in each cell happens to be an integer and the same for each cell, but, in general, the density is not an integer and differs between cells with different race labels. Fig 1D shows an alternative representation of the pattern seen in Fig 1C; each cell has a race label and a population density value.

The RL disambiguation of inhabitants' locations within a block is stochastic. Repeating disambiguation multiple times results in an ensemble of patterns that differ in details of their spatial configuration but preserve composition. Because the RL is constructed using the smallest available census subdivisions (census blocks) the values of segregation and diversity metrics are not affected by the stochastic nature of the disambiguation procedure. This is because census blocks are highly homogeneous. In order to verify this statement quantitatively we calculated 50 realizations for each county in a sample of 51 counties having different spatio-racial patterns ($MI$ ranges from 0.007 to 0.375 and $E$ ranges from 0.91 to 2.13). The goal was to observe the spread (quantified by a standard deviation) of the values of segregation and diversity metrics in each county due to differences in spatial configurations between realizations. The spread is 0.0002-0.002 for $MI$ and 0.0003—0.005 for $E$ depending on the county. This demonstrates that, to the good approximation, the values of $MI$ and $E$ do not change from one realization to another. For further explanations and the data, see the section Racial Landscapes at https://socscape.edu.pl.

The RL does not improve the spatial resolution of census block data; it artificially disambiguates the spatial location of racial sub-populations within a block. The upside of this transformation is that, in its new format, the data forms a continuous pattern across the entire region of interest (ROI). This format supports pattern-based, zoneless calculation of the segregation metric.

Transforming census data from all blocks in the ROI results in a large raster of the type shown in Fig 1D. Technically, there are two rasters, one categorical storing race labels, and another numerical storing population densities. These two rasters are used for both

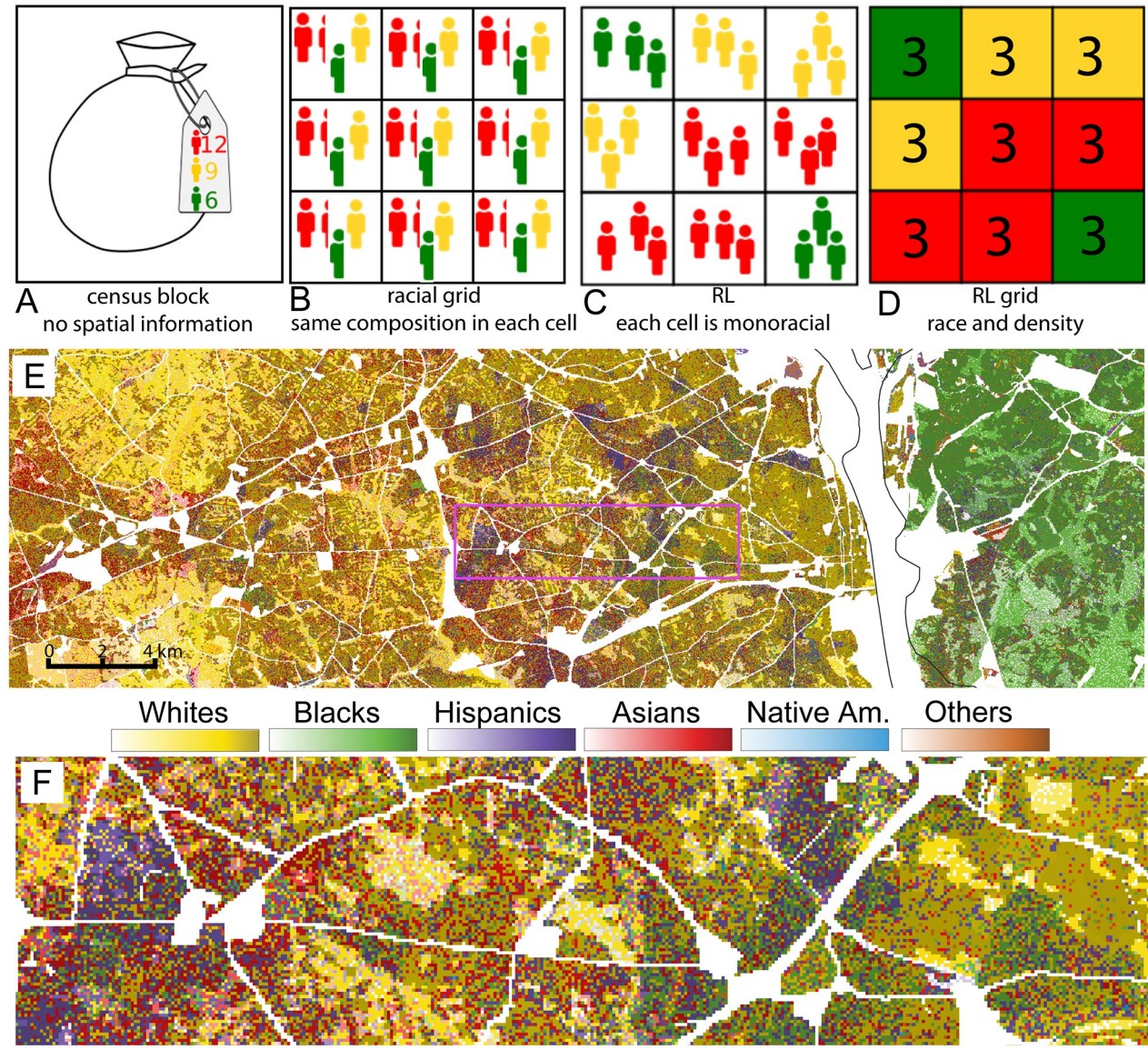

**Fig 1. Illustration of the Racial Landscape (RL) method.** A-D: Data transformation from a census block (A) to the RL grid (D). In this illustration, all cells in (D) have the same densities, but in general, densities vary. Different colors reflect different racial sub-populations. E: An example of the RL image showing the racial geography of parts of Washington, DC. Each cell has one of six colors corresponding to six contributing racial sub-populations, but same-race cells may differ in shades, depicting population density—the darker the shade, the larger the local population density. F: The RL image of the area enclosed in the purple rectangle. At this magnification, the details of population distribution begin to emerge. Uninhabited areas are shown in white.

visualization (mapping racial geography) and zoneless calculation of the segregation metric (see the next sub-section).

Using race as a color and density as a shade of this color (see the legend to Fig 1), the RL constructs an RL-image, a bird's-eye-like view visualization of racial geography. It can also be thought of as a racial map. Fig 1E shows the RL image for a portion of the Washington, DC area located just south/west of the National Airport (DCA). This image provides a comprehensive visualization of the racial geography with a high level of detail. Fig 1F zooms in on a

specific region bounded by a purple rectangle in Fig 1E, further highlighting the level of detail captured by the RL image. These examples clearly demonstrate the ability of the RL methodology to visualize racial geography with exceptional detail.

## Racial Landscape: Calculating metric of racial segregation and diversity

In the RL, the segregation metric is calculated from a pattern formed by cells in the RL raster. This is analogous but not identical to how an aggregation metric is calculated in the pattern of land cover [28]. In Fig 2A, a hypothetical RL raster is shown, with three race categories shown by three colors. Numbers represent population densities per cell. The categorical part of the raster has the same form as land cover data. In such a case, the summary of aggregation is encapsulated by a co-occurrence matrix [29]. However, whereas in land cover aggregation is assessed by area, in racial demographics, segregation is assessed by population counts. Thus, the exposure matrix is a co-occurrence matrix weighted by local population density (Fig 2B). For example, a categorical co-occurrence between green and red races is 3 (using rook cells' adjacency), but by incorporating local population densities, a numerical exposure between green and red races is 8.5 (see Fig 2C).

Normalizing the exposure matrix (Fig 2B) through division by the sum of all its entries yields a joint probability distribution function $p(x, y)$ where $x$ and $y$ are people's races in two adjacent cells. For example, $p(red, green) = 0.18$ is a probability of drawing red and green inhabitants from adjacent cells. High probabilities of drawing same-color inhabitants and low probabilities of drawing different-color inhabitants indicate high segregation. In general, the segregation metric is given by the so-called mutual information (see Dmowska et al. [26] for details). There are two forms of mutual information: absolute (denoted by $MI$) and normalized by entropy (denoted by $NMI$). Entropy is a measure of population diversity (see below).

The $NMI$ metric plays the same role as the $H$ index in traditional analysis; it has a range between 0 and 1 and indicates the level of segregation regardless of ROI's diversity. The $NMI$ should be used to compare the segregation of different cities that may have different diversities. The $MI$ metric plays the same role as the numerator in the formula for $H$ (see, for example, Iceland [30]); it depends on the region's diversity. The monoracial ROI is scored high by $NMI$ (assessed as segregated), but is scored low by $MI$ (assessed as not segregated).

The RL calculates a racial diversity index as an entropy $E$ of the normalized racial composition of an ROI. This is no different from how the diversity of a multiracial population has been assessed in traditional methodology [3, 4], except that the RL, being a raster dataset, allows for calculating entropy from a region of any size and shape. To provide immediate interpretability to the diversity metric, we transform $E$ into a quantity known as Hill's number ($N_H = a^E$), where $a$ represents the base of the logarithm used to calculate the marginal entropy (in this case, $a = 2$). Hill's number [31] indicates the number of distinct races that significantly contribute to the overall population of the ROI. Note that $N_H$ may not be an integer, and it should be rounded to obtain the number of different races present. The NRGD2020 lists values of $N_H$ and $MI$, and values of entropy can be obtained from $E = \log_2 N_H$, and values of $NMI$ can be obtained from $NMI = MI/\log_2 N_H$.

The unique feature of RL is that it connects quantitative analysis (values of diversity and segregation metrics) with spatial visualization (a racial map). Fig 2D to 2F provide examples of such connection. They show racial maps (RL images) of three different ROIs together with numerical assessments of their diversity and segregation. Visual examination of the maps indicates that the region shown in Fig 2E is the most diverse and the region shown in Fig 2F is the least diverse. Similarly, just by looking at the maps, it is clear that the region shown in Fig 2D is the most segregated and the region shown in Fig 2E is the least segregated. All these

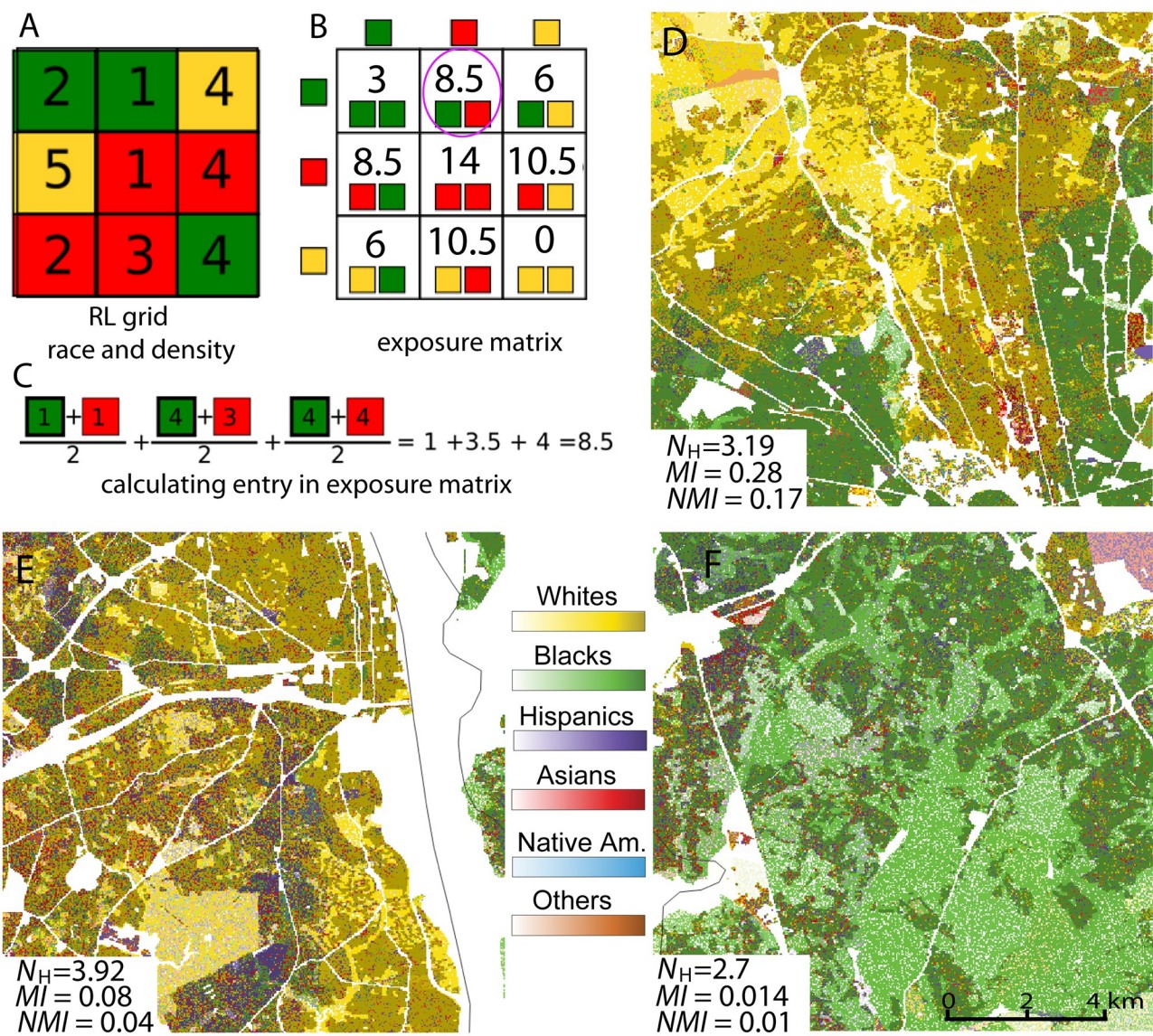

**Fig 2. Calculating a metric of racial segregation using the RL method.** A: RL grid with each cell having a given race and storing the value of local population density. B: An exposure matrix calculated from RL grid layers. The exposure matrix is a co-occurrence matrix weighted by an average of local population densities in neighboring cells. C: An example of calculating an entry to the exposure matrix using the red-green (Black-Asian) cells. D-F: Examples of RL image visualizations of three 12km × 12km areas in Washington DC and values of $N_H$, $MI$, and $NMI$ calculated from them.

observations are supported by the metrics. Thus, using the RL image, an analyst can visually assess the character of racial geography even before any quantitative analysis is done.

## Computation of NRGD2020

NRGD2020 is a collection of high-resolution geospatial layers calculated using the RL. It uses 2020 census data and covers the entire conterminous US. NRGD2020 makes it possible to take advantage of the unique features of the RL without performing extensive computations.

Fig 3 depicts the outline of the procedure we employed to calculate the layers for NRGD2020. The numbered circles serve as reference points for the description. The input

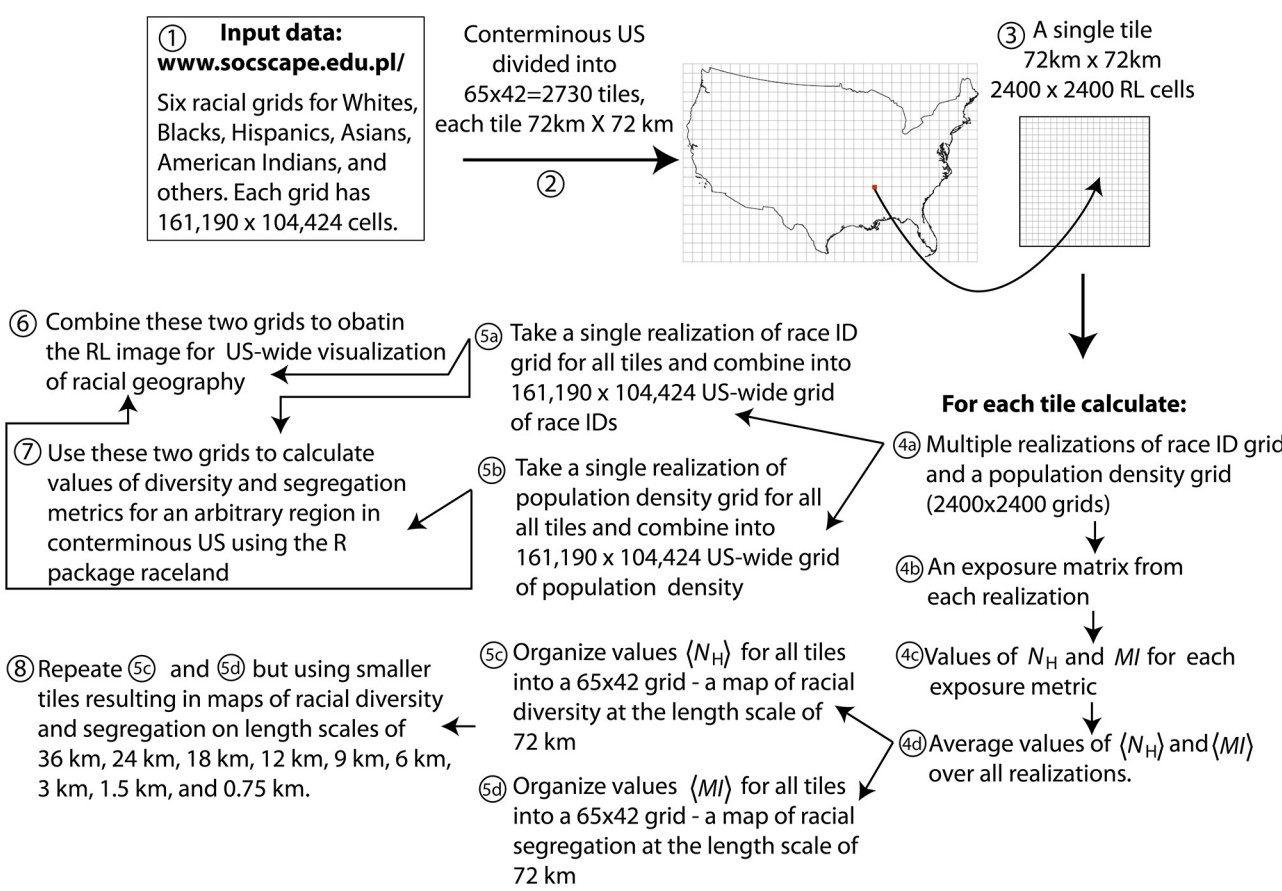

**Fig 3. Outline of our procedure to calculate the National Racial Geography Dataset 2020.** Numbers are pointers used in the main text.

data ① consists of six racial grids, comprising 30-meter resolution rasters that cover the entire conterminous US. Each cell within these rasters contains the density value for a specific sub-population, including Whites, Blacks, Hispanics, Asians, American Indians, and others, as of 2020. For the sake of brevity, we refer to each sub-population as a "race," although we acknowledge that the Hispanic category represents ethnicity.

The stochastic nature of the RL methodology results in an extensive computer memory requirement. To alleviate this requirement in ②, we divide the racial grids into non-overlapping tiles of size $2400 \times 2400$ cells (72 km × 72 km). One such tile is shown in ③. In ④a, we calculate fifty realizations of the racial pattern within each tile. Steps ④b to ④d are performed to calculate ensemble averages of metrics $N_H$ and $MI$ for each tile.

Next, the procedure forks into two branches. The first branch starts at ④a. In step ⑤a, a single realization of a pattern of racial IDs for the entire US is constructed from individual tiles. Similarly, in step ⑤b, a US-wide numerical grid of cell-based population density is created from individual tiles. In ⑥, these two grids are used to create the RL image, and in ⑦, they are used to create a US-wide race/density grid for extracting an arbitrary ROI for further analysis, including the calculation of diversity and segregation metrics using the R package *raceland* (https://cran.r-project.org).

The second branch of the fork starts at ④a. In step ⑤c, a US-wide map of spatial variability of diversity at the scale of 72 km is created, and in ⑤d, a US-wide map of racial segregation at

**Table 1. The list of layers in the NRGD2020.**

| Layer | resolution | description | use |
|---|---|---|---|
| RL image | 30-meters | An RGB image that provides a US-wide visualization of racial geography | Visual analysis. |
| Diversity grids | 72-km, 36-km, 24-km, 18-km, 12-km, 9-km, 6-km, 3-km, 1.5-km, 0.75-km | Rasters showing spatial variability of $N_H$ over the conterminous US at ten different length scales. | To visualize and quantify dependence of racial diversity on length scale. |
| Segregation grids | 72-km, 36-km, 24-km, 18-km, 12-km, 9-km, 6-km, 3-km, 1.5-km, 0.75-km | Rasters showing spatial variability of $MI$ over the conterminous US at ten different length scales. | To visualize and quantify dependence of racial segregation on length scale. |
| RL racial ID grid | 30-meters | A raster in which each cell has a label corresponding to one of six races. | This layer is the first necessary input for the R package *raceland* which calculates metrics of segregation and diversity for an arbitrary area. It can also be used to visualize racial pattern without taking population density into consideration. |
| RL population density grid | 30-meters | A raster in which each cell has a numerical value indication local population density. | This layer is a second necessary input for the R package *raceland* which calculates metrics of segregation and diversity for an arbitrary area. |

the scale of 72 km is created. In ⑧, analogous variability maps are created for nine additional scales.

## Results

Geospatial layers constituting the NRGD2020 dataset are listed and summarized in Table 1. These layers are provided in the GeoTIFF format and can be downloaded from https://socscape.edu.pl.

The first layer in Table 1 is the RL image, which provides a high-resolution US-wide map of racial geography. Maps displayed in Figs 1 and 2 were extracted from this layer. Each cell's (pixel's) color represents a race, and the color's shade represents population density; a darker shade depicts larger density. As seen from a legend shown in Figs 1 and 2, cells with negligible density are depicted by whitish tones, irrespective of the race. When examining the map, one's attention is initially drawn to the number of different colors, providing a visual assessment of racial diversity. Additionally, the degree of mixing between different colors offers insights into the level of local racial segregation.

When visually assessing segregation using the RL image, it is essential to keep in mind that the assessment pertains to the segregation within the mapped area, rather than the segregation of the mapped area from the larger region around it. Quantitative assessment of segregation can be performed visually from an RL map.

A region depicted by coarse patterns of colors is strongly segregated. This is evident in the region shown in Fig 2D, which is divided into yellow (symbolizing Whites) and green (symbolizing Blacks) sectors. The shade of yellow in the yellow sector varies spatially according to variations in population density, but all cells in this sector contain only White inhabitants (with some exceptions). Similarly, the green part contains only cells inhabited by Black inhabitants. Also note that there are numerous individual red cells within the yellow part of the region, indicating the presence of Asian inhabitants. Thus, there is more than just two (Black and White) racial groups with a significant presence in this region, as its value of $N_H \approx 3$ instead of 2. Its value of $MI = 0.28$; for an interpretation of $MI$ values, see Dmowska et al. [26].

Region depicted by a fine pattern of colors is weakly segregated. This is the situation in the region shown in Fig 2E. This region is also highly diverse, as cells of four different colors are

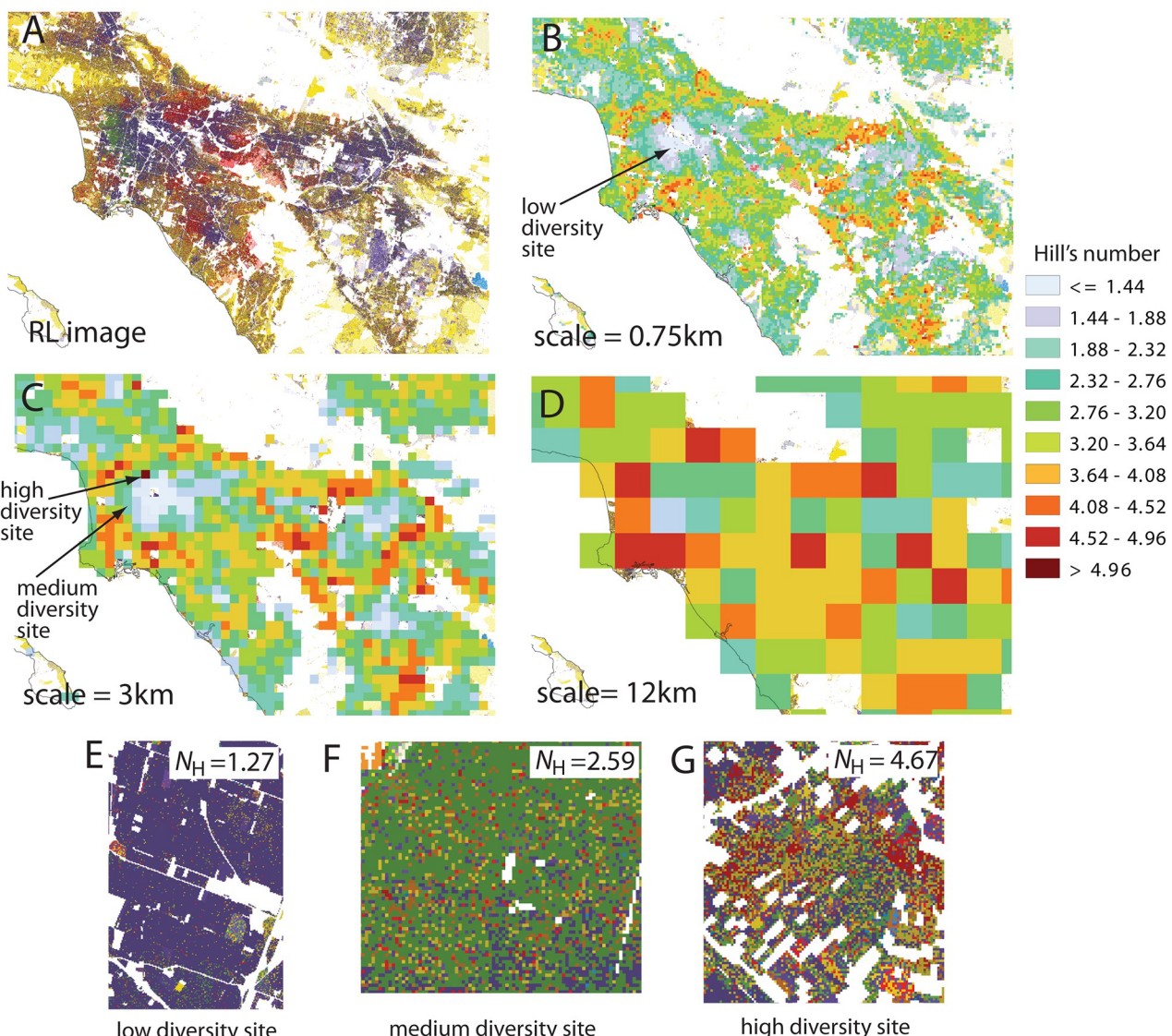

**Fig 4. An example of mapping racial diversity at different length scales.** A: The RL image of the Los Angeles area serves as the reference map. B-D: Racial diversity maps at length scales of 0.75 km, 3 km, and 12 km, respectively, overlaying the RL image. E-G: Examples of three locations in the Los Angeles area exhibiting low, medium, and high racial diversity, respectively. The pointers indicate the mapping of these locations on the diversity maps.

observed in significant numbers. This observation is supported by the values $N_H \approx 4$ and $MI = 0.08$.

Examination of a region shown in Fig 2F reveals that four different sub-populations are present, but the Black sub-population is dominant. Moreover, while green cells (representing the Black sub-population) are spatially aggregated, cells of other colors (representing three other sub-populations) are spatially disaggregated. Thus, a visual assessment of this region indicates a medium level of diversity and a low level of segregation. The segregation is low because the dominant sub-population is spatially aggregated while other sub-populations are not. This is confirmed by the values $N_H \approx 3$ and $MI = 0.01$.

The subsequent two layers listed in Table 1 are precalculated US-wide maps of diversity and segregation at different scales. Fig 4 provides an illustration of the visual appearance and

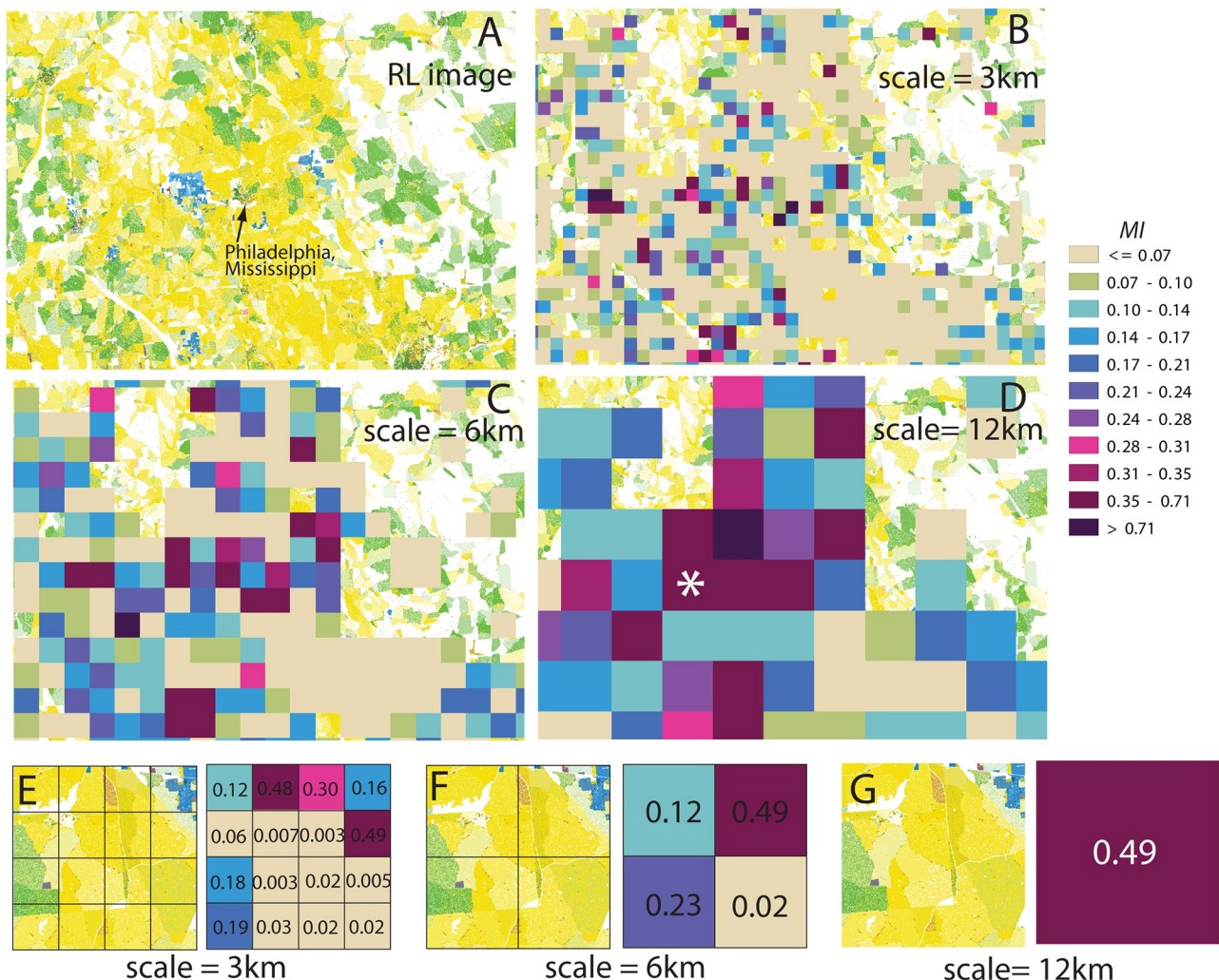

**Fig 5. An example of mapping racial segregation at different length scales.** A: The RL image of the rural area in Mississippi (referred to as the "Choctaw site") serves as the reference map. B-D: Racial segregation maps at length scales of 3 km, 6 km, and 12 km, respectively, overlying the RL image. E-G: Segregation maps at the three length scales for a single 12 km × 12 km sub-tile marked by a star symbol on panel (D).

interpretation of precalculated diversity layers using Los Angeles (LA) as an example. Fig 4A displays the racial geography of LA, clearly indicating the significant presence of sub-populations such as Hispanics and Asians, in addition to Whites and Blacks. Fig 4B to 4D present precalculated diversity maps calculated at increasing scales of 0.75 km, 3 km, and 12 km, respectively. These maps demonstrate the continuity of diversity across different scales, as observed through the resemblance between the 3 km scale map (Fig 4C) and the 0.75 km scale map (Fig 4B), as well as the similarity between the 12 km scale map (Fig 4D) and the 3 km scale map (Fig 4C). Fig 4E to 4G showcase RL maps at specific locations, marked by pointers, to visually validate the corresponding values of the $N_H$ metric. For instance, Fig 4E exhibits a nearly homogeneous Hispanic population, which aligns with an $N_H$ value of 1.27.

Fig 5 provides an illustration of the visual appearance and interpretation of precalculated segregation layers, using a rural area in Mississippi as an example. This particular site is centered around the small city of Philadelphia, with a population of approximately 7,000, and is

located near the Choctaw Indian reservation. Fig 5A displays the racial geography of the site, revealing a segregated mixture of White (yellow color) and Black (green color) residential areas, along with areas inhabited by American Indians (blue color). Fig 5B to 5D depict segregation maps calculated at increasing scales of 3 km, 6 km, and 12 km, respectively.

In Fig 5B (scale 3 km), segregated square sub-tiles are observed along the boundaries between White, Black, and American Indian zones, while the sub-tiles within these zones appear monoracial and thus not segregated (represented by the beige color). As the scale increases, the association between segregated sub-tiles and zone boundaries becomes less apparent, as the size of the tiles approaches the granularity of the racial pattern. Fig 5E to 5G showcase a specific 12 km × 12 km sub-tile marked by a star symbol in Fig 5D. At the 12 km scale, this sub-tile has a segregation metric *MI* value of 0.49 (Fig 5G). Upon further dividing the sub-tile into smaller sub-tiles (Fig 5E and 5F), variations in local segregation values become apparent. At the smallest scale of 3 km, most sub-tiles are completely within the White zone, resulting in low *MI* values indicative of the absence of segregation.

While the first three layers listed in Table 1 are precalculated and ready to use for analysis without any additional computations, the last two layers listed in Table 1 are input data for further processing. They are intended for analytical tasks for which precalculated layers are not sufficient. These layers are to be utilized for computing metrics of diversity ($N_H$) and segregation (*MI*) for a user-defined ROI using the R package *raceland*. Additional details regarding these two layers will be provided below in a "Use cases" section.

## Use cases

In this section, we illustrate how the NRGD2020 dataset, along with the R package *raceland*, can be employed to address racial geography problems similar to those examined in existing literature. We present two examples here. Our first example compares racial diversity and segregation within metropolitan statistical areas; it utilizes RL racial ID grid and RL population density grid (fourth and fifth entries in Table 1) and requires the use of the R package *raceland*. In the second example, we show the construction of segregation profiles; it utilizes only precalculated segregation grids (third entry in Table 1).

### Comparing racial geographies of different parts of metropolitan statistical area (MSA)

A common inquiry among racial demographers involves comparing various zones within metropolitan statistical areas (MSAs) in terms of racial diversity and segregation. For example, a recent study by Lichter et al. [32] utilized census data from 1990 to 2020 to analyze such differences and their temporal changes across multiple MSAs throughout the United States.

For each MSA, they classified the constituent counties into four distinct zones: principal cities, inner-ring suburbs, outlying suburbs, and fringe suburbs. Subsequently, they calculated the values of diversity metrics (specifically Simpson's Diversity Index) and segregation metrics (specifically the Index of Dissimilarity) for each zone, employing the division of zones into census tracts or a place (for Atlanta principal city). Their choice of the segregation metric was driven by their focus on segregation between specific racial pairs (such as Black-White), rather than multigroup segregation.

Here, we demonstrate how to conduct a similar analysis focusing on multigroup segregation using the NRGD2020 dataset and the R package *raceland*. Our case study pertains to the Atlanta metropolitan statistical area (MSA) in 2020. To begin, we crop the RL grid racial ID and RL grid population density (as listed in Table 1) to the boundaries of the Atlanta MSA region. Fig 6 showcases the RL image, which serves as a visualization layer illustrating the two

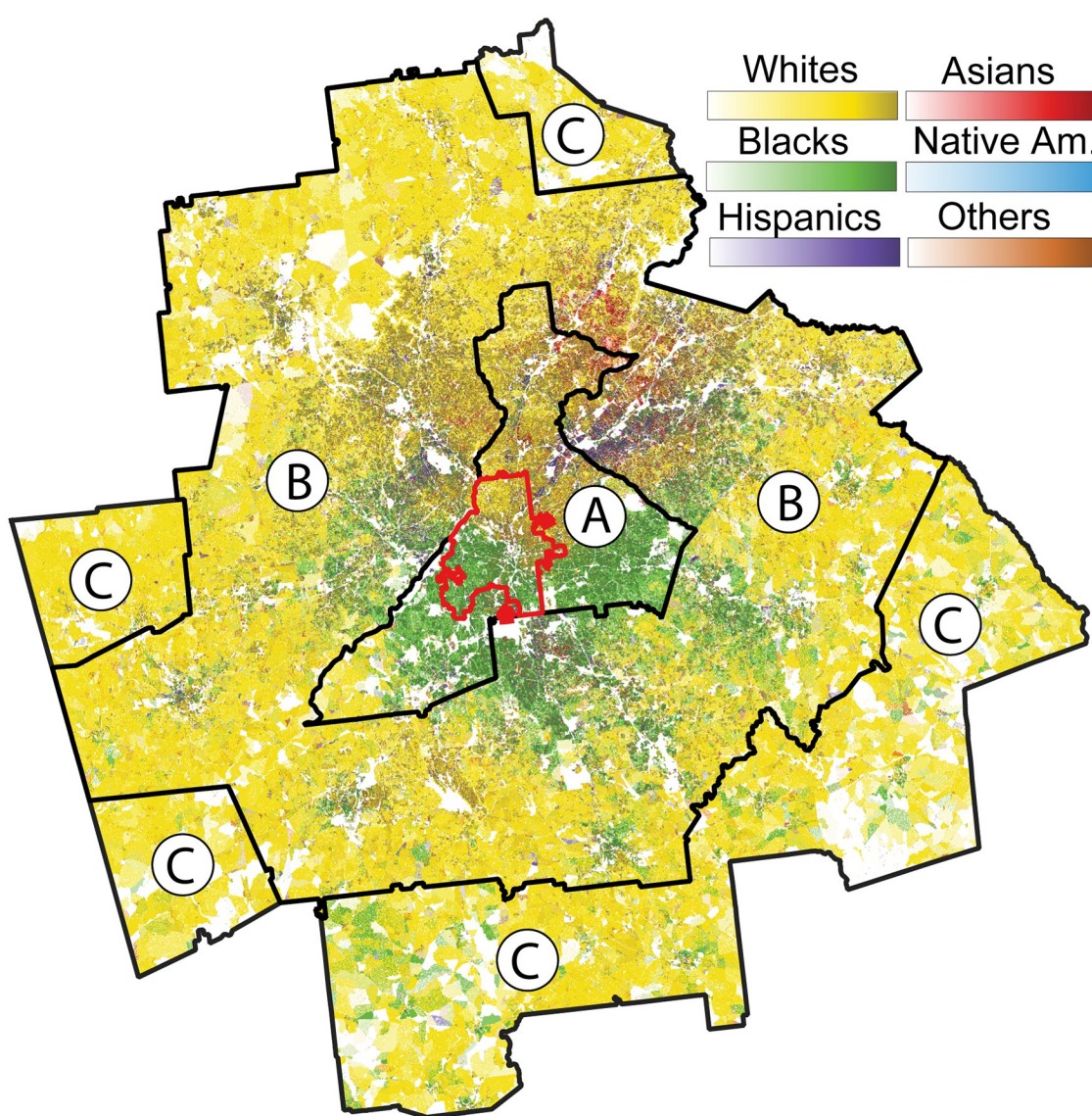

| Zone | $E$ | $N_H$ | $MI$ | $NMI$ |
|---|---|---|---|---|
| MSA | 1.88 | 3.68 | 0.193 | 0.103 |
| principal city | 1.69 | 3.23 | 0.258 | 0.153 |
| inner-ring subs. (A) | 1.87 | 3.66 | 0.316 | 0.169 |
| outlying subs. (B) | 1.89 | 3.71 | 0.135 | 0.071 |
| fringe subs. (C) | 1.09 | 2.13 | 0.123 | 0.113 |

**Fig 6. RL image of Atlanta MSA overlaid by the boundaries of the four zones.** The red boundary outlines the principal city, black boundaries outline three types of suburbs, inner-ring (A), outlying (B), and fringe (C). The table lists values of diversity and segregation metrics for the zones.

RL grid layers, specifically limited to the Atlanta MSA. Additionally, the boundaries defining the four zones established by Lichter et al. [32] are superimposed on the RL image. Subsequently, we utilize the boundaries of the Atlanta census "place" (for the principal city zone) and of the counties constituting the MSA to define the boundaries of the three suburban zones. Finally, we execute the *raceland* code for each zone to compute values of metrics of diversity ($N_H$) and segregation ($MI$) for each zone. These values are listed in the table included in Fig 6.

## Segregation profiles

Another frequent inquiry among racial demographers pertains to the appropriate scale at which segregation should be measured. Traditionally, in racial demography, the term "scale" refers to the size of local subdivisions within a region for which segregation is calculated. Consequently, the level of segregation depends on how these local subdivisions are defined. To address this issue, Reardon et al. [33] introduced the concept of the segregation profile, which represents a function describing the segregation level across different scales. This profile was established by computing the segregation of a region multiple times, utilizing the information theory index $H$ [3, 4], while employing local circular subdivisions with varying radii. It is worth noting that segregation profiles are typically monotonic decreasing functions of scale (radii of local subdivisions). This is due to the fact that racial diversity within local areas tends to increase as their size grows, resulting in a lower value of $H$ as per its definition.

We will now demonstrate the process of obtaining a segregation profile using the precalculated segregation grids available in the NRGD2020 dataset. Note that our segregation profile differs conceptually from those introduced by Reardon et al. [33]. This distinction arises from the fact that the RL method allows for the calculation of segregation for arbitrary areas. Therefore, our segregation profile represents a function of the segregation level based on the size of the region, rather than a function of the segregation of the region based on the size of its subdivisions.

For this demonstration, we have selected six different MSAs: Jackson, MS; New Orleans, LA; Chicago, IL; New York, NY; Los Angeles, CA; and Knoxville, TN. We begin by cropping the US-wide segregation grids to the boundaries of the six chosen MSAs. Recall that each cell in these grids contains the segregation metric values ($MI$) calculated for the region within that cell. Thus, for each MSA, we obtain the segregation levels at different scales (corresponding to varying sizes of localities) by simply averaging the values of the cells within each scale-specific segregation grid.

Fig 7B consists of a sequence of maps showcasing the RL image of the Jackson MSA, followed by three NRGD2020 precalculated segregation maps at length scales of 3, 9, and 18 km, respectively. Similarly, Fig 7C exhibits the corresponding content but for the Chicago MSA. Fig 7A presents the six segregation profiles.

Note that all segregation profiles demonstrate a monotonic increase as a function of the length scale, as expected. This trend arises because larger tiles have the potential for greater segregation. Additionally, it is worth mentioning that the Jackson MSA exhibits higher levels of segregation compared to the Chicago MSA across all length scales. This result is visually supported by the series of segregation maps depicted in Fig 7B and 7C, where the proportion of monoracial tiles is consistently higher in Jackson compared to Chicago at each scale.

The R code and dataset to replicate both examples are available from the NRGD2020 section of the https://socscape.edu.pl website.

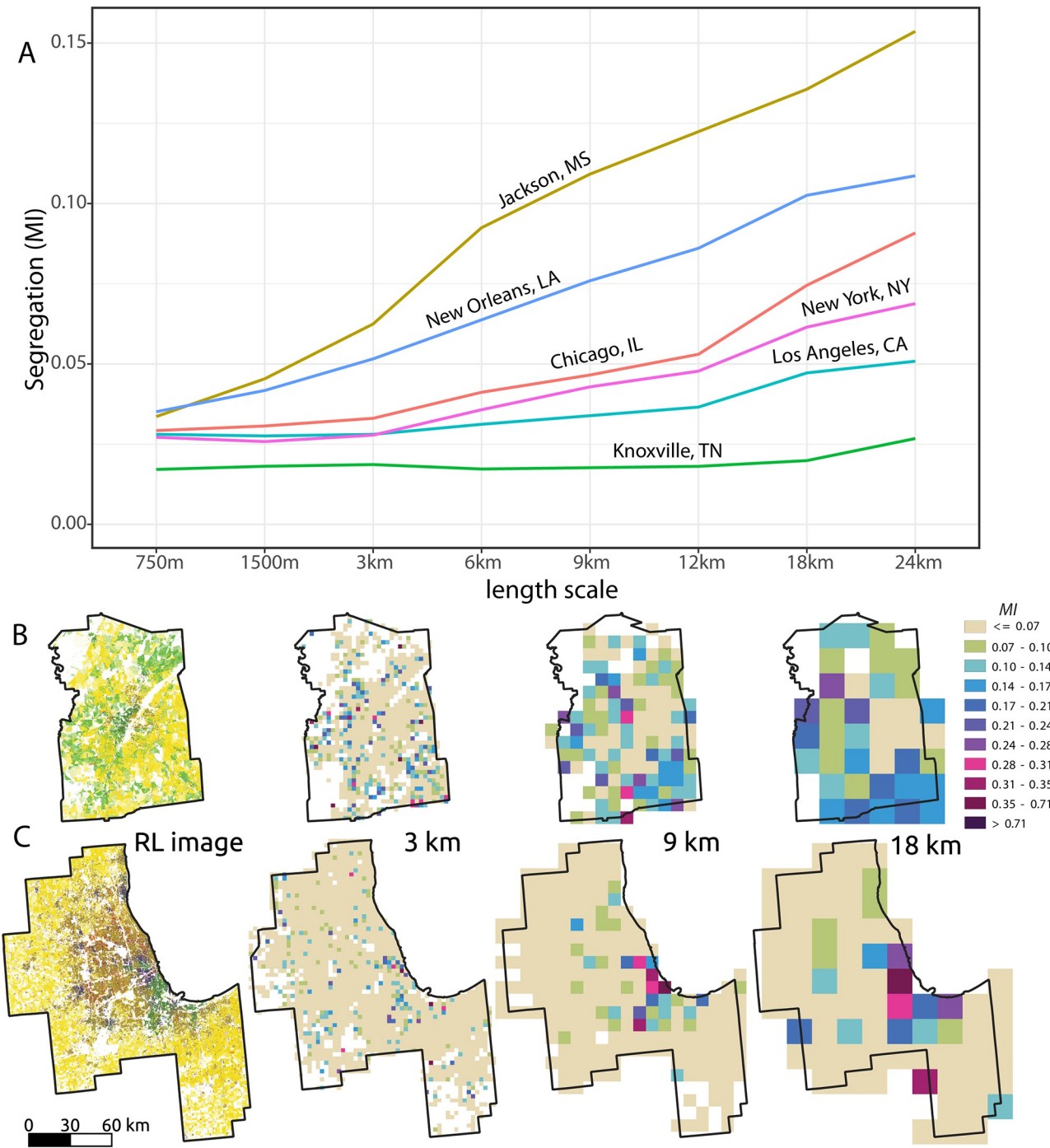

**Fig 7. Changes in segregation level within a scale in six MSAs.** A: Segregation profiles for six MSAs, see main text for details. B: A sequence of maps showcasing the Jackson MSA, RL image followed by three NRGD2020 precalculated segregation maps at length scales of 3, 9, and 18 km, respectively. C: The same content as (B) but for the Chicago MSA.

**Table 2. Comparison of traditional approach and RL method in analyzing and visualizing racial geography.**

| Feature | Traditional method | RL method |
|---|---|---|
| Segregation metrics | Values of metrics depend on census subdivisions. | Divisions are not used for calculation of segregation metrics. |
| Region of interests | Segregation can be calculated only for a MSA, a county, or a census tract. | Segregation can be calculated for any region of interests. |
| Multiscale segregation | Segregation is calculated for a MSA or a county using a series of subdivisions of different granularity. This provides an indirect and not-mappable measure of segregation scales. | Region is partitioned into a series of lattices of different granularity and segregation is calculated for every tile in a lattice to produce a multiresolution series of maps of segregation variability over the ROI. This provides a direct and mappable information on segregation scales. |
| Connecting quantification and visualization of racial geography | Segregation can be visualized by mapping values of *H* using census tracts or larger granularity. | Values of segregation and diversity metrics can be examined vis-à-vis the RL image. |

## Discussion

The aim of this paper was to introduce a dataset (NRGD2020) which expedites and extends spatial analysis of racial geography. NRGD2020 covers the entire conterminous US in 2020. There are two types of layers in the NRGD2020, precalculated layers, which are ready for analysis without any additional computing, and data layers, which provide input to the R package *raceland* to compute numerical assessment of racial segregation and diversity for a region of arbitrary size and shape. The NRGD2020 is meant to ease an adoption of the Racial Landscape methodology of analyzing racial geography by enabling its use without a need for a deeper technical understanding of RL's stochastic methodology.

Because of its RL roots, performing spatial analysis of racial geography with NRGD2020 does not require acquisition of census data and shapefiles. Methodologically, the biggest differentiation between conventional analysis and NRGD2020-based analysis is that the later can be used to assess segregation and diversity of an arbitrary region without worrying about census boundaries and divisions. This enables doing types of analyses that are not possible with conventional methods. In addition, all NRGD2020-based quantitative analysis can be examined "vis-à-vis" the RL image visualization (see Figs 2 and 4 to 7). A consistency between quantitative results and RL image increases a confidence in the validity of the assessment and understanding of analysis.

To summarize the advantages of using RL and NRGD2020 over traditional methods of analysis and census-aggregated data, Table 2 compares the traditional method to the Racial Landscape in terms of their abilities to assess segregation, choose the region of interest, analyze multiscale segregation, and visualize racial geography. In the rest of the discussion, we will comment on entries in Table 2.

### Segregation metrics

The RL's zoneless calculation of a racial segregation metric is likely to be the most puzzling feature. This is because calculating segregation index from census aggregates is a deeply entrenched concept in racial demography. However, from a geographical viewpoint, assessing the strength of a spatial phenomenon like racial residential segregation from aggregated data does not seem to be the most natural approach. Instead, from the geographic perspective, a distribution of multiracial population is a continuous spatial pattern and the phenomenon of the segregation is tantamount to the clumping of that pattern. The RL is a tool that converts census aggregated data into a continuous spatial pattern while accommodating spatial uncertainty within a block.

Another potentially puzzling aspect of the RL is its stochastic construction. To convert aggregated data to a grid, the RL stochastically organizes a population within each census block into sub-block size monoracial cells. This does not reduce uncertainty about within the block's locations of inhabitants, but the inhabitants are now organized into a continuous spatial pattern (even if arbitrarily) instead of being all mixed in one bag (as illustrated in Fig 1A). Thus, the RL grid for the entire region is stochastic—each realization of the RL grid is different every time it is generated. However, these differences are only present at the scale of a census block, at larger scales they smooth out. Indeed, a previous study [26] has demonstrated that the standard deviations of diversity and segregation metrics calculated from an ensemble of patterns realizations are very small at scales of analysis larger than a typical size of a census block (see Table 2 in Dmowska et al. [26]). Thus, it is alright to use a stochastic pattern to represent spatial distribution of multiracial population at scales larger than a size of a block.

Does a segregation metric calculated from a clumping of the racial pattern measures segregation in the same sense as the index $H$? The answer is yes. As we pointed out in "Materials and methods" section the metric $NMI$ measures segregation in the same sense as $H$ although the numerical values of the two assessments differ. The empirical study by Dmowska and Stepinski [34] has shown that values of $H$ and values of $NMI$ calculated for a set of fifty-one major MSAs have a rank correlation of 0.92. Thus, comparing segregation in different MSAs using $H$ and $NMI$ leads to very similar conclusions. However, $NMI$ can be used to assess segregation for a region of arbitrary size and shape, while $H$ can only assess segregation of census aggregates divided into smaller census aggregates.

## Region of interests

A distinctive feature setting apart the RL-NRGD2020 method from traditional approaches lies in its capability to compute segregation within an arbitrarily defined ROI. This makes possible the assessment of segregation at various scales, whether within a zip code, school district, congressional district, or any user-defined area. Additionally, it facilitates the partitioning of a large region into a lattice of tiles of arbitrary size $L$ offering the means to map the spatial variability of segregation across the entire region. This functionality stems from the method's foundation, quantifying segregation as the degree of clumping within a pattern.

## Scale of segregation

An important question in racial geography analysis pertains to the characteristic length of segregated areas. The significance of racial segregation is often heightened when it occurs on a larger scale, as observed in many cities in the Midwest US. While each region exhibits a spectrum of segregation scales, there might be a dominant scale at which most of the population is segregated. For instance, in a region depicted in Fig 2D, segregation between Black and White groups occurs on a large scale, whereas in the region shown in Fig 2E, segregation between different racial groups transpires on a smaller scale.

In the existing literature, the assessment of segregation scale is accomplished through multiscale segregation analysis. This entails calculating segregation for a region (e.g., a MSA) using various subdivisions of different granularity (for an overview see Owen et al. [35]). Since census aggregates provide only up to three granularities (tracts, block groups, and blocks), some authors have employed alternative divisions based on physical size (e.g., [33, 36]) or population count (e.g., [18, 37]). As granularity becomes finer, the segregation index $H$ tends to increase. However, the growth of $H$ may not follow a linear pattern, with bumps in the growth curve indicating significant scales of segregation. This indirect approach helps assess characteristic scales of segregation.

The RL provides a direct approach to this task. Fig 5E to 5G illustrate this approach. The left parts of these three panels show that this $12 \times 12$ km region is segregated on the large scale (relative to its size). The region is divided into three lattices with granularities 3, 6, and 12 km. The *MI* index is calculated for each tile in each granularity. The largest average growth of tile's *MI* with increased granularity is from 3 km to 6 km. Thus, the dominant scale of segregation is 6 km, which is in agreement with what is seen in the RL image. NRGD2020 provides layers precalculated for the 10 different granularities showing how segregation (and, separately, diversity) varies spatially over the conterminous US.

## Mapping racial geography

Visualization of a dataset often provides a broader understanding of the data that quantitative analysis may lack. Given that racial segregation is a spatial phenomenon, the obvious choice for visualizing racial data is through mapping. However, racial data in the form of census aggregates does not lend itself easily to informative mapping, especially for segregation. Maps of neighborhood classifications [14, 16, 19] may be interpreted in terms of segregation, even if they do not explicitly map segregation. On the other hand, the RL image maps segregation on all scales (see examples in Figs 1 and 2), and NRGD2020's precalculated layers, listed in row three of Table 1, show quantitative maps of spatial variability of racial segregation (see Fig 5) on different scales.

In summary, our proposed approach provides a comprehensive solution that enables seamless visualization and quantitative analysis through the integration of the NRGD2020 dataset and the *raceland* package. Despite the limitations imposed by the US census, the RL image offers the closest representation of the true spatial distribution of racial sub-populations. These accurate and information-rich maps of racial geography bring real-world context to quantitative demographic analysis. To conduct quantitative analysis, including the calculation of diversity and segregation metrics, the ROI data extracted from NRGD2020 can be directly inputted into the R package *raceland*, as demonstrated in Section "Use cases".

Moving forward, the next step will be to construct NRGDs for the years 1990, 2000, and 2010. This collection of NRGDs for different census years would open up new possibilities for studying the temporal changes in racial geography.

## Author Contributions

**Conceptualization:** Anna Dmowska, Tomasz F. Stepinski.

**Data curation:** Anna Dmowska.

**Formal analysis:** Anna Dmowska.

**Funding acquisition:** Anna Dmowska, Tomasz F. Stepinski.

**Investigation:** Anna Dmowska, Tomasz F. Stepinski.

**Methodology:** Anna Dmowska, Tomasz F. Stepinski.

**Resources:** Anna Dmowska.

**Visualization:** Anna Dmowska, Tomasz F. Stepinski.

**Writing – original draft:** Tomasz F. Stepinski.

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
