## [Decision Letter · Decision Letter 0]

10 Dec 2023

PONE-D-23-31167Enhancing the quantification and mapping of racial geography in the United States using the National Racial Geography DatasetPLOS ONE

Dear Dr. Dmowska,

Thank you for submitting your manuscript to PLOS ONE. After careful consideration, we feel that it has merit but does not fully meet PLOS ONE’s publication criteria as it currently stands. Therefore, we invite you to submit a revised version of the manuscript that addresses the points raised during the review process. Both the qualified reviewers think your paper addresses an important and interesting question but raise some concerns about the data accuracy and other issues. The second reviewer is indifferent between recommending "major revision" and "rejection," but I think your paper aims to document your new method and new data output  and this deserves publication. Please try to address the reviewers' concerns or provide rebuttal  if you are unable to address some. In addition, I find your statement “the existing literature primarily focuses on quantifying residential diversity and segregation within large metropolitan areas” is true only for public data. The restricted use of census data contain geographic information down to the block level for each individual so theoretically racial diversity can be calculated precisely at the block level using the confidential census data.

We look forward to receiving your revised manuscript.

Kind regards,

Shihe Fu, Ph.D.

Academic Editor

PLOS ONE

2. We note that Figures 1E, 1F, 2D, 2E, 2F, 3, 4A, 4B, 4E, 4F, 4G, 5, 6, 7B, and 7C. in your submission contain [map/satellite] images which may be copyrighted. All PLOS content is published under the Creative Commons Attribution License (CC BY 4.0), which means that the manuscript, images, and Supporting Information files will be freely available online, and any third party is permitted to access, download, copy, distribute, and use these materials in any way, even commercially, with proper attribution. For these reasons, we cannot publish previously copyrighted maps or satellite images created using proprietary data, such as Google software (Google Maps, Street View, and Earth). For more information, see our copyright guidelines: http://journals.plos.org/plosone/s/licenses-and-copyright.

1. You may seek permission from the original copyright holder of Figures 1E, 1F, 2D, 2E, 2F, 3, 4A, 4B, 4E, 4F, 4G, 5, 6, 7B, and 7C to publish the content specifically under the CC BY 4.0 license. 

Reviewers' comments:

Reviewer's Responses to Questions

**Comments to the Author**

1. Is the manuscript technically sound, and do the data support the conclusions?

Reviewer #1: Yes

Reviewer #2: No

2. Has the statistical analysis been performed appropriately and rigorously? 

Reviewer #1: No

Reviewer #2: Yes

3. Have the authors made all data underlying the findings in their manuscript fully available?

Reviewer #1: Yes

Reviewer #2: Yes

4. Is the manuscript presented in an intelligible fashion and written in standard English?

Reviewer #1: Yes

Reviewer #2: No

5. Review Comments to the Author

Reviewer #1: This research aims to present a fine-grained dataset of racial demography for the U.S. in 2020. Due to privacy concerns, the existing studies generally measure racial segregation at the level of census tract, which is less accurate. Based on their previous research on racial landscapes, the authors transform vector-based census data into high-resolution grid data. This dataset potentially promotes the development of racial demography. However, there are several major issues needed to be addressed.

1. Current metrics for racial diversity and segregation should be examined. The authors need to engage with the extant literature concerning proxies of racial demography, compare different metrics and justify the choice of metrics of this research.

2. Accuracy of this dataset should be tested carefully. What is the accuracy of this dataset? Does the accuracy vary spatially? If so, why? Due to the stochastic nature of this approach, how does the number of realization affect the accuracy of the dataset? Does the dataset have any accuracy gain compared to other datasets/approaches?

Some minor points:

3. Three metrics (NH, MI, and NMI) should be compared in a table. What are their strengths and weaknesses? What is the relationship between each metric and segregation/diversity (i.e., racial spatial pattern)? Are they mutually reinforcing?

4. There should be more discussion of the limitations of the dataset.

Reviewer #2: The aim of this study was to improve the accessibility and effectiveness of 435 racial geography studies within the GIS environment. The authors sought to convert vector-based census data into high-resolution raster data and present a fine-grained dataset of racial demographics for the United States in 2020. This is a very interesting and important topic to discuss, but I do not think the argument is entirely new. Again, this is probably not entirely new and can certainly be found in the existing literature, but it is worth collecting in one place along with the above point. While I recognize the roles of this paper, I am not sure its contribution is original enough for the PLOS ONE standard.

1, Theoretically, the authors argue that recent studies expose racial geography in the emergent context of the U.S. as a new apparatus of regionalism, However, this dichotomy is not convincing, as regionalism and the role of the market and the state are intensely linked and debated in the literature.

2, As for the literature review, perhaps more attention should be paid to this in the introduction, as I found the existing literature strangely under-cited. Particularly in the literature review, these types of sources, the racial demography and regionalism literature are cited more extensively.

3, Data: The accuracy of the data set should be justified, how do the authors intend to convince their peers of this data? The reasons for choosing this data set should be carefully explained.

4, Method: The methods used to convert vector-based census data into high-resolution raster data in this study should be explained and discussed, what are the advantages of the method chosen in this study?

4, Discussion and conclusion: the conclusion should address the research questions and deal with the current literature, which are the basic points of the research paper. The discussion section should also discuss the limitations of this research and its methods.

5, the English language should be checked more carefully as there are numerous English grammatical and typographical errors in the paper.

6. PLOS authors have the option to publish the peer review history of their article (what does this mean?). If published, this will include your full peer review and any attached files.

Reviewer #1: No

Reviewer #2: No

---

## [Author Response · Author response to Decision Letter 0]

25 Jan 2024

Considering the comments of both reviewers, we significantly rewrote the paper, including the changes to the title. We also wrote a new Discussion section of the paper that focuses on descibing the differences between analyzing and visualizing segregation and diversity using traditional approaches and our dataset with the RL methodology. We hope this new version clearly states the novelty of our approach compared to the traditionally used methods. 

Editor comments 

In addition, I find your statement “the existing literature primarily focuses on quantifying residential diversity and segregation within large metropolitan areas” is true only for public data. The restricted use of census data contain geographic information down to the block level for each individual so theoretically racial diversity can be calculated precisely at the block level using the confidential census data.

As the editor mentions, the individual-level data are confidential and have restricted access. Thus, it cannot be commonly used in racial diversity and segregation analysis. The lack of individual-level data prompts the researcher to develop methods that can be applied to the census aggregated data. The existing literature focuses on analyzing segregation and diversity using census-aggregated data that are openly available.

Reviewer 1

Reviewer #1: This research aims to present a fine-grained dataset of racial demography for the U.S. in 2020. Due to privacy concerns, the existing studies generally measure racial segregation at the level of census tract, which is less accurate. Based on their previous research on racial landscapes, the authors transform vector-based census data into high-resolution grid data. This dataset potentially promotes the development of racial demography. However, there are several major issues needed to be addressed.

We would like to make several points here. 

(1) The grid used in the present paper is monoracial, meaning each cell groups inhabitants of a single race only. This is in contrast to previous grids that are all multiracial.

(2) The high resolution of the grid has been a distinct feature of previous multiracial grids; here, the high resolution is a secondary feature, and monoraciallity is the primary feature.

(3) This paper does not introduce a monoracial grid (it has already been introduced). Instead, it introduces the NRGD dataset that allows researchers to calculate the segregation of an arbitrary region within the conterminous US without the technical knowledge needed to generate a monoracial grid. Of course, the present paper contains a brief on the RL method (a method of generating a monoracial grid and using it for the zoneless calculation of segregation). Still, it is a non-technical brief that focuses on key concepts.

1. Current metrics for racial diversity and segregation should be examined. The authors need to engage with the extant literature concerning proxies of racial demography, compare different metrics and justify the choice of metrics of this research.

The novelty here is not a better metric of segregation but the ability to calculate segregation for an arbitrarily defined region of interest (ROI) not restricted to a census aggregated area divided into smaller census aggregated areas – a restriction of the traditional method. The need for such functionality is self-evident, but we still justify it in several places in the revised paper. To have such functionality, the segregation metric used in the paper has to adhere to the transformed data in order to work due to the need for a “new” metric. Thus, a review of existing segregation metrics would be out of topic in this paper as they do not apply to the monoracial grid. The revised paper also states that using a set of 50 US metropolitan areas, the rank correlation between the metric NMI used to assess segregation from a pattern and the index H used in traditional calculation assessment is 0.92. Thus, NMI measures segregation in the same sense as H but is not restricted to census aggregated areas. 

2. Accuracy of this dataset should be tested carefully. What is the accuracy of this dataset? Does the accuracy vary spatially? If so, why? Due to the stochastic nature of this approach, how does the number of realization affect the accuracy of the dataset? Does the dataset have any accuracy gain compared to other datasets/approaches?

The question needs to be more specific. The term “accuracy of the dataset” is too broad as the dataset consists of many layers representing different variables. The accuracy of transformed data, the monoracial grid, is, by design, the same as the accuracy of census block data; integrating all cells in the block yields the racial composition as listed by Census block-level data. In other words, a sub-census spatial distribution is arbitrary, but this is OK, as our goal was not to increase spatial accuracy below the census block but to change the structure of the data so the segregation metric can be calculated from a pattern. This is stated in the revised paper.

The accuracy of the segregation metric is also limited by the accuracy of census block data. See our reply to the next question on the issue of how a stochastic approach influences the accuracy of the segregation metric. 

Due to the stochastic nature of this approach, how does the number of realization affect the accuracy of the dataset?

As in all stochastic approaches, an ensemble mean is more accurate (the standard deviation is smaller) for a larger number of realizations. This issue has been discussed in the technical paper on the RL methodology. (Dmowska et al. 2020, https://www.sciencedirect.com/science/article/abs/pii/S0143622819310367). The result is that accuracy is more sensitive to the size of the area over which the value of the segregation metric is computed than the number of realizations. We determined that with 50 realizations, the accuracy of the metric for areas larger than a census block is very high.

To illustrate to the reviewer that the number of realizations does not affect the results, we chose 50 counties and, for each county, calculated 50 realizations. In the figure below, the black points present the segregation metrics calculated for each realization; the red point is a mean calculated from 50 realizations. It can be observed that the deviation of metrics values is indeed small regardless of segregation metrics values.

Some minor points:

3. Three metrics (NH, MI, and NMI) should be compared in a table. What are their strengths and weaknesses? What is the relationship between each metric and segregation/diversity (i.e., racial spatial pattern)? Are they mutually reinforcing?

In the revised version of the paper, we include a separate subsection in the Material and Methods that describes and explains those 3 metrics. They are not measuring the same property, so their comparison in a table would not provide any more information. NH measures diversity; it is just an entropy expressed in an easier-to-interpret manner. MI measures a degree of clumping of the same-race cells in a pattern; clumping of racial pattern corresponds to racial segregation. By its design, MI increases with the increase in the number of different racial sub-populations. Thus, according to MI, a completely segregated city with three significant races is more segregated than a completely segregated city with two races. NMI is the MI relative to the diversity of the region measured by the entropy (NMI=MI/E = MI/(Log NH) ). Thus, according to NMI, a completely segregated city with three significant races has the same level of segregation as a completely segregated city with two races. This is explained in the revised paper.

4. There should be more discussion of the limitations of the dataset.

The limitations of the RL method and the limitation of the NRGD dataset are two different issues. Both RL and NRGD are introduced to remove limitations of the traditional method and census aggregated data, respectively; therefore, by design, they have fewer limitations than the traditional approach.

Keeping in mind that the primary quantification goal of RL is to enable assessing segregation for an arbitrary region, the only limitation (stated in the paper) is that the size of the region needs to be larger than a census block. The limitation of the dataset is self-explanatory from the list in Table 1. The pre-calculated grids for local diversity and segregation (rows 2 and 3 in the revised Table 1) are calculated only for square areas at 10 different scales. In reality, this is not a significant limitation. To calculate segregation and diversity of arbitrary area, which is not pre-calculated, a user has to choose an ROI from race and density grids (rows 4 and 5 in the revised table 1) and do calculations in the R package raceland. Is it a limitation? We don’t think so because the package is freely available and relatively easy to use. 

REVIEWER 2

Reviewer #2: The aim of this study was to improve the accessibility and effectiveness of 435 racial geography studies within the GIS environment. The authors sought to convert vector-based census data into high-resolution raster data and present a fine-grained dataset of racial demographics for the United States in 2020. This is a very interesting and important topic to discuss, but I do not think the argument is entirely new. Again, this is probably not entirely new and can certainly be found in the existing literature, but it is worth collecting in one place along with the above point. While I recognize the roles of this paper, I am not sure its contribution is original enough for the PLOS ONE standard.

The work presented in this paper is entirely new. Although descriptions of quantitative methods of assessing racial segregation abound in the literature, they all restricted to census-aggregated areas divided into other census-aggregated areas. Thus, a traditional method can assess the segregation of a county or a metropolitan statistical area but not, for example, a zip code or a congressional district.

Only recently was the Racial Landscape (RL) method published that allows for assessing segregation of an area of arbitrary size and shape. The RL is unlike any previous methods as it uses a stochastic approach to transform census block racial data into sub-block cells of monoracial cells, thus converting racial data to a spatial pattern from which a zoneless assessment of segregation is possible. This stochastic underpinning of RL, unfamiliar to the demographic community, presents a barrier to the adaption of RL in racial studies, thus missing the opportunity to perform studies not possible under the traditional framework. The present paper aims to introduce the NRGD dataset, which contains multiple layers of information pre-calculated using the RL for the entire conterminous US in 2020. This makes it possible to take advantage of the RL without necessarily a technical understanding of its stochastic underpinning.

1, Theoretically, the authors argue that recent studies expose racial geography in the emergent context of the U.S. as a new apparatus of regionalism, However, this dichotomy is not convincing, as regionalism and the role of the market and the state are intensely linked and debated in the literature.

We don’t understand this remark, but certainly, it has no relation to the core of our paper. Likely, the reviewer refers to the introduction, where we describe some regional differences in the distribution of different races. This was done only to point out that the spatial distribution of different racial populations is not simple. This description is no longer present in the revised manuscript to avoid confusion.

2, As for the literature review, perhaps more attention should be paid to this in the introduction, as I found the existing literature strangely under-cited. Particularly in the literature review, these types of sources, the racial demography and regionalism literature are cited more extensively.

As we have already explained in our answer to the first question, the methodology we work with has no technical connection to methodologies described in past literature. Thus, a review of past literature would be out of place. Furthermore, the RL method itself is not a focus of this paper (although we give a non-technical brief on the RL method). It was a focus of an earlier paper (Dmowska et al. 2020, https://www.sciencedirect.com/science/article/abs/pii/S0143622819310367) where several citations were given in which standards methods of quantification of racial segregation are reviewed. In this paper, we focus on the construction of the NRGD dataset and the demonstration of its usefulness for tasks for which the standard methodology is not adequate.

3, Data: The accuracy of the data set should be justified, how do the authors intend to convince their peers of this data? The reasons for choosing this data set should be carefully explained.

The reason for choosing racial data represented as a monoracial grid is clearly explained in the paper (especially in the revised manuscript) and has nothing to do with accuracy. The reason is that a monoracial grid (a spatial pattern) is the form of racial data supporting zoneless segregation calculation, while previously used aggregated multiracial format does not. The two use cases presented in the paper clearly illustrate the advantage of the ability to calculate segregation for arbitrary areas, including square blocks.

As for the actual accuracy of monoracial gridded data, by design, it is the same as the accuracy of census block data. Integrating all cells in the block yields the racial composition as listed by census block-level data. Redistribution of block inhabitants to sub-block cells is not to increase the accuracy but to form a pattern.

4, Method: The methods used to convert vector-based census data into high-resolution raster data in this study should be explained and discussed, what are the advantages of the method chosen in this study?

As we already explained, a technical description of RL construction is not the topic of this paper. The full description is available in the previous paper (Dmowska et al. 2020, https://www.sciencedirect.com/science/article/abs/pii/S0143622819310367).

We have already explained that the ability to calculate the segregation of an arbitrary area without using any zones is the key advantage of the method. The additional advantage is the tight integration of visualization and quantification of racial geography.

4, Discussion and conclusion: the conclusion should address the research questions and deal with the current literature, which are the basic points of the research paper. The discussion section should also discuss the limitations of this research and its methods.

This is a kind of paper that does not have a specific research question. Such papers are explicitly allowed by PlosOne. The purpose of the paper is to introduce an extensive dataset that would expedite and improve studies in which the calculation of racial segregation anywhere in the US and on any scale is needed.

In the revised paper, we rewrote the Discussion section. It now focuses on describing the comparison between standard methods and our approach. 

5, the English language should be checked more carefully as there are numerous English grammatical and typographical errors in the paper.

We check the English in the revised version.

---

## [Decision Letter · Decision Letter 1]

27 Mar 2024

PONE-D-23-31167R1Quantification and visualization of US racial geography using the National Racial Geography Dataset 2020PLOS ONE

Dear Dr. Dmowska,

Thank you for submitting your manuscript to PLOS ONE. After careful consideration, we feel that it has merit but does not fully meet PLOS ONE’s publication criteria as it currently stands. Therefore, we invite you to submit a revised version of the manuscript that addresses the points raised during the review process.

The second reviewer did not provide comments; the first reviewer has one minor concern. In order not to delay the publication of your paper, based on my own reading, I invite you for a minor revision. It is up to you how much you can or would like to address the first reviewer's concern. Your revised version will not be sent out for refereeing and will be accepted directly.

We look forward to receiving your revised manuscript.

Kind regards,

Shihe Fu, Ph.D.

Academic Editor

PLOS ONE

Journal Requirements:

Reviewers' comments:

Reviewer's Responses to Questions

**Comments to the Author**

1. If the authors have adequately addressed your comments raised in a previous round of review and you feel that this manuscript is now acceptable for publication, you may indicate that here to bypass the “Comments to the Author” section, enter your conflict of interest statement in the “Confidential to Editor” section, and submit your "Accept" recommendation.

Reviewer #1: (No Response)

2. Is the manuscript technically sound, and do the data support the conclusions?

Reviewer #1: Yes

3. Has the statistical analysis been performed appropriately and rigorously? 

Reviewer #1: Yes

4. Have the authors made all data underlying the findings in their manuscript fully available?

Reviewer #1: Yes

5. Is the manuscript presented in an intelligible fashion and written in standard English?

Reviewer #1: Yes

6. Review Comments to the Author

Reviewer #1: The authors largely addressed my previous concerns. However, there is one remaining issue. In response to my query regarding the impact of the stochastic nature of their approach on the accuracy of the dataset, the authors demonstrated that the number of realizations barely affects the mutual information (i.e., the segregation metrics). Despite this clarification, the effect of the number of realizations on diversity metrics remains unexplored and warrants further investigation. Although this topic was briefly mentioned in section 2.4 of their paper published in Applied Geography (https://www.sciencedirect.com/science/article/abs/pii/S0143622819310367), the discussion lacked comprehensive detail. Therefore, a more thorough analysis of the stochastic characteristics of RL and its implications for the dataset should be incorporated into this manuscript. This inclusion will not only enhance the completeness of the research but also provide a clearer understanding of how the number of realizations influences the broader outcomes of the study.

7. PLOS authors have the option to publish the peer review history of their article (what does this mean?). If published, this will include your full peer review and any attached files.

Reviewer #1: No

---

## [Author Response · Author response to Decision Letter 1]

14 Apr 2024

Reviewer #1: The authors largely addressed my previous concerns. However, there is one remaining issue. In response to my query regarding the impact of the stochastic nature of their approach on the accuracy of the dataset, the authors demonstrated that the number of realizations barely affects the mutual information (i.e., the segregation metrics). Despite this clarification, the effect of the number of realizations on diversity metrics remains unexplored and warrants further investigation. Although this topic was briefly mentioned in section 2.4 of their paper published in Applied Geography (https://www.sciencedirect.com/science/article/abs/pii/S0143622819310367), the discussion lacked comprehensive detail. Therefore, a more thorough analysis of the stochastic characteristics of RL and its implications for the dataset should be incorporated into this manuscript. This inclusion will not only enhance the completeness of the research but also provide a clearer understanding of how the number of realizations influences the broader outcomes of the study.

As it is mentioned by the reviewer, we clarify that the number of realizations does not affect the segregation metrics. The same is true about the racial diversity metric. We perform test using 51 counties characterized by different spatio-racial pattern, and the results of this test are available in section Racial Landscapes at https://socscape.edu.pl In the revised manuscript we added the lines 92-106 to clarify this issue, and for further explanation we included also link to the https://socscape.edu.pl/index.php?id=racial-landscapes 

The conclusions from our test are as follows: 

 • In our sample of 51 counties the mean segregation metric MI calculated using 50 realizations change from 0.007 to 0.379 (with the standard deviation between 0.0002-0.002), and the mean diversity metric E range from 0.91 to 2.13 (with the standard deviation between 0.0003 - 0.005).

 • In Figure 1 (please see the document at https://socscape.edu.pl/index.php?id=racial-landscapes ) the black points present the segregation metrics calculated for each of realization; the red point is a mean calculated from 50 realizations. The difference between mutual information values (black point) is so small that is comparable to the symbol size in this plot (symbol size is 0.5 mm), for some counties the black points are covered by the red point (mean value). The conclusions from this plot is that the deviation of segregation metrics values is small regardless of segregation metrics values. 

 • In Figure 2 (please see the document at https://socscape.edu.pl/index.php?id=racial-landscapes ) the black points present the diversity metrics calculated for each realization; the red point is a mean calculated from 50 realizations. It can be observed that the deviation of metrics values are even smaller than for the segregation metric MI. 

 • The Spearman correlation between segregation metric calculated using NRGD (1 realization), and the mean values from 5, 10, 20, 30, 40, and 50 realizations range from 0,9997 to 0,9999. It indicates that the rankings based on segregation metrics are almost identical. The further investigation shows that 2 pairs of counties switch their position.

 • For example, comparing the rankings based on segregation metrics calculated using 50 realizations, and 1 realization derived from NRGD:

 ◦ counties Shelby, AL and San Francisco, CA switch the positions between 15 and 16. (Shelby, AL: MI based on 50 realizations is equal to 0.44835, and based on NRGD is 0.04534; San Francisco, CA: MI based on 50 realizations is equal to 0.04490, and based on NRGD is 0.04512)

 ◦ counties Bronx, NY and Kent, MI switch their positions between 22 and 23. (Kent, MI: MI based on 50 realizations is equal to 0.08036, and based on NRGD is 0.0796; Bronx, NY: MI based on 50 realizations is equal to 0.08013, and based on NRGD is 0.08039)

 • The Spearman correlation between diversity metric calculated using NRGD, and the mean values from 5, 10, 20, 30, 40, and 50 realizations range from 0,9996 to 0,9997.

 • For example, comparing the rankings based on diversity metrics calculated using 50 realizations, and 1 realization derived from NRGD:

 ◦ counties Bronx, NY and Bermallillo, NM switch the positions between 25 and 26. (Bronx, NY: E based on 50 realizations is equal to 1.6878, and based on NRGD is 1.6838; Bermallillo, NM: E based on 50 realizations is equal to 1.6849, and based on NRGD is 1.6843)

 ◦ counties Santa Clara, CA and San Diego, CA switch the positions between 33 and 34. (Santa Clara, CA : E based on 50 realizations is equal to 1.8795, and based on NRGD is 1.8776; San Diego: E based on 50 realizations is equal to 1.8791, and based on NRGD is 1.8784)

---

## [Decision Letter · Decision Letter 2]

11 Jul 2024

Quantification and visualization of US racial geography using the National Racial Geography Dataset 2020

PONE-D-23-31167R2

Dear Dr. Dmowska,

We’re pleased to inform you that your manuscript has been judged scientifically suitable for publication and will be formally accepted for publication once it meets all outstanding technical requirements.

Kind regards,

Abdulkader Murad, Ph.D

Academic Editor

PLOS ONE

Additional Editor Comments (optional):

Reviewers' comments:

Reviewer's Responses to Questions

**Comments to the Author**

1. If the authors have adequately addressed your comments raised in a previous round of review and you feel that this manuscript is now acceptable for publication, you may indicate that here to bypass the “Comments to the Author” section, enter your conflict of interest statement in the “Confidential to Editor” section, and submit your "Accept" recommendation.

Reviewer #1: All comments have been addressed

2. Is the manuscript technically sound, and do the data support the conclusions?

Reviewer #1: Yes

3. Has the statistical analysis been performed appropriately and rigorously? 

Reviewer #1: Yes

4. Have the authors made all data underlying the findings in their manuscript fully available?

Reviewer #1: Yes

5. Is the manuscript presented in an intelligible fashion and written in standard English?

Reviewer #1: Yes

6. Review Comments to the Author

Reviewer #1: (No Response)

7. PLOS authors have the option to publish the peer review history of their article (what does this mean?). If published, this will include your full peer review and any attached files.

Reviewer #1: No

---

## [Editor Report · Acceptance letter]

16 Jul 2024

PONE-D-23-31167R2 

PLOS ONE

Dear Dr. Dmowska, 

I'm pleased to inform you that your manuscript has been deemed suitable for publication in PLOS ONE. Congratulations! Your manuscript is now being handed over to our production team.

Kind regards, 

on behalf of

Professor Abdulkader Murad 

Academic Editor

PLOS ONE